# The Genetic Variants Influencing Hypertension Prevalence Based on the Risk of Insulin Resistance as Assessed Using the Metabolic Score for Insulin Resistance (METS-IR)

**DOI:** 10.3390/ijms252312690

**Published:** 2024-11-26

**Authors:** Bo-Kyung Shine, Ja-Eun Choi, Young-Jin Park, Kyung-Won Hong

**Affiliations:** 1Department of Family Medicine, Medical Center, Dong-A University, Busan 49201, Republic of Korea; kdwsbk@naver.com; 2Institute of Advanced Technology, Theragen Health Co., Ltd., Seongnam 13493, Republic of Korea; jaeun.cho@theragenhealth.com

**Keywords:** METS-IR, hypertension, GWAS, SNP, KoGES

## Abstract

Insulin resistance is a major indicator of cardiovascular diseases, including hypertension. The Metabolic Score for Insulin Resistance (METS-IR) offers a simplified and cost-effective way to evaluate insulin resistance. This study aimed to identify genetic variants associated with the prevalence of hypertension stratified by METS-IR score levels. Data from the Korean Genome and Epidemiology Study (KoGES) were analyzed. The METS-IR was calculated using the following formula: ln [(2 × fasting blood glucose (FBG) + triglycerides (TG)) × body mass index (BMI)]/ ln [high-density lipoprotein cholesterol (HDL-C)]. The participants were divided into tertiles 1 (T1) and 3 (T3) based on their METS-IR scores. Genome-wide association studies (GWAS) were performed for hypertensive cases and non-hypertensive controls within these tertile groups using logistic regression adjusted for age, sex, and lifestyle factors. Among the METS-IR tertile groups, 3517 of the 19,774 participants (17.8%) at T1 had hypertension, whereas 8653 of the 20,374 participants (42.5%) at T3 had hypertension. A total of 113 single-nucleotide polymorphisms (SNPs) reached the GWAS significance threshold (*p* < 5 × 10^−8^) in at least one tertile group, mapping to six distinct genetic loci. Notably, four loci, rs11899121 (chr2p24), rs7556898 (chr2q24.3), rs17249754 (ATP2B1), and rs1980854 (chr20p12.2), were significantly associated with hypertension in the high-METS-score group (T3). rs10857147 (FGF5) was significant in both the T1 and T3 groups, whereas rs671 (ALDH2) was significant only in the T1 group. The GWASs identified six genetic loci significantly associated with hypertension, with distinct patterns across METS-IR tertiles, highlighting the role of metabolic context in genetic susceptibility. These findings underscore critical genetic factors influencing hypertension prevalence and provide insights into the metabolic–genetic interplay underlying this condition.

## 1. Introduction

Hypertension is a significant global health issue, recognized as a primary risk factor for mortality and a major contributor to various cardiovascular diseases (CVDs), including stroke and ischemic heart disease [1,2]. According to the World Health Organization (WHO), approximately one-third of the adult population worldwide is affected by hypertension, accounting for around 51% of strokes and 45% of deaths from ischemic heart disease attributable to elevated blood pressure [3,4]. In response to these alarming statistics, research is increasingly focusing on the underlying pathophysiology of hypertension. This involves exploring several mechanisms, including autacoids, hormonal regulation, genetic predisposition, oxidative stress, inflammation, and alterations in vascular function [5].

Insulin resistance and compensatory hyperinsulinemia are critical mechanisms involved in the development of essential hypertension [6,7,8]. Hyperinsulinemia, resulting from insulin resistance, elevates blood pressure by stimulating the sympathetic nervous system and increasing sodium reabsorption in the renal tubules [5]. Numerous studies conducted over the past few decades have confirmed a significant association between insulin resistance and hypertension [9,10,11,12,13,14,15,16,17].

Moreover, hypertension is closely linked to metabolic syndrome [18,19], a condition characterized by a cluster of metabolic irregularities, such as insulin resistance, obesity, dyslipidemia, and hyperglycemia [20,21,22,23]. The coexistence of hypertension and metabolic syndrome significantly increases the risk of cardiovascular disease [24,25,26]. However, it is important to note that not all patients with hypertension present with metabolic syndrome, and conversely, not all individuals with metabolic syndrome develop hypertension. This complexity suggests that the heterogeneous nature of hypertension development is influenced by genetic factors, indicating that personalized treatment strategies that consider genetic background may improve hypertension management [5].

Recent studies have underscored the pivotal role of genetic factors in the pathogenesis and severity of hypertension [27,28], a trend that is mirrored in its association with metabolic syndrome [29]. In particular, the Metabolic Score for Insulin Resistance (METS IR), a key indicator of metabolic syndrome [30,31,32], serves as a reflection of insulin resistance and may help explain the disparities in the prevalence of hypertension between the high- and low-METS IR groups [33]. In light of these findings, this study conducted a large-scale genome-wide association study (GWAS) focusing on hypertension within populations stratified by METS-IR levels and identified genetic variants associated with hypertension. The outcomes of this study are intended to identify crucial biomarkers for predicting hypertension risk and providing treatment guidelines, thereby enhancing our understanding of the intricate interactions between hypertension and metabolic syndrome.

The primary objective of this study is to elucidate the genetic associations underlying hypertension prevalence. By identifying key biomarkers associated with hypertension, this research aims to enhance predictive capabilities and inform personalized treatment approaches. Furthermore, the findings provide valuable insights into the complex pathophysiological interactions between hypertension and metabolic syndrome, supporting the development of tailored therapeutic strategies for affected individuals.

## 2. Results

### 2.1. Clinical Characteristics

We conducted a GWAS using the large-scale Korean Epidemiology Cohort dataset, KoGES, and the overall research process is summarized in the study design illustration in Figure 1. Table 1 presents the demographic and clinical characteristics of the participants stratified by METS-IR score tertiles. The division into tertiles was chosen to balance granularity and statistical power, ensuring sufficient differentiation among low-, moderate-, and high-metabolic-risk groups while maintaining robust sample sizes. Participants in the highest tertile (T3) were older and had a higher BMI, waist circumference, fasting glucose levels, and triglycerides and lower HDL-C levels than those in the lowest tertile (T1). Among the METS-IR tertile groups, 3517 of the 19,774 participants (17.8%) at T1 had hypertension, whereas 8653 of the 20,374 participants (42.5%) at T3 had hypertension. The prevalence of hypertension was significantly higher in the T3 group (*p* < 0.001).

### 2.2. Genome-Wide Association Study

We conducted a GWAS for prevalence of hypertension in both the METS-IR T1 and T3 groups. The statistical significance of the GWAS results was determined using a genome-wide significance *p*-value threshold of 5 × 10^−8^. The SNPs that passed this threshold were classified into clusters within the same locus based on their linkage disequilibrium (LD) relationships, as detailed in Appendix A, which also includes references from previous GWASs of the corresponding loci. To facilitate the comparison of the results between the T1 and T3 groups, the overall GWAS results were visualized using a Miami plot, as shown in Figure 2. Significant SNP loci are highlighted with boxes, and the major mapped gene names are indicated. As a result, a total of six loci associated with the prevalence of hypertension were identified.

The SNPs with the highest number of reports and significant *p*-values for each locus were selected as the lead SNPs and are summarized in Table 2.

We identified six significant genetic markers associated with the prevalence of hypertension. Among these, three markers, rs11899121 on chromosome 2p24, rs7556898 on chromosome 2q24.3, and rs17249754 in the ATP2B1 gene, lowered the prevalence of hypertension in individuals with a high METS-IR. Additionally, the rs671 marker in the ALDH2 gene was associated with reduced prevalence of hypertension risk in individuals with a low METS-IR. Another marker, rs1980854, on chromosome 20p12.2, was associated with an increased prevalence of hypertension in individuals with a high METS-IR. Lastly, the rs10857147 marker in the FGF5 gene increased the prevalence of hypertension irrespective of METS-IR levels.

## 3. Discussion

In the present study, we identified six significant genetic markers associated with the prevalence of hypertension. Notably, four loci (rs11899121 [chr2p24], rs7556898 [chr2q24.3], rs17249754 [ATP2B1], and rs1980854 [chr20p12.2]) were significantly associated with hypertension in the high-METS-IR group (T3), rs10857147 (FGF5) was significant in both T1 and T3, and rs671 (ALDH2) was significant only in T1. These findings suggest that both METS-IR and genetic predisposition play crucial roles in modulating the prevalence of developing hypertension, highlighting the need for a multifaceted approach to understanding and managing hypertension.

Of the six loci identified, rs7556898 and rs1980854 are novel findings that have not been previously reported in GWASs, representing unique contributions to the genetic understanding of hypertension. The rs11899121 marker has been reported in earlier studies but was not previously associated with hypertension in the context of METS-IR stratification, emphasizing the relevance of metabolic risk stratification in uncovering new associations. Markers in ATP2B1 [34], ALDH2 [35,36], and FGF5 [37] have been reported in previous studies, corroborating their roles in blood pressure regulation and cardiovascular health. These novel and replicated findings expand the current understanding of the genetic–metabolic interplay in hypertension, suggesting that metabolic stratification can reveal hidden genetic associations that might otherwise remain undetected in broader populations. Further replication studies are warranted to validate these findings and elucidate their biological mechanisms.

It is worth noting that the participants in this study were all of Korean ethnicity. Korea is well known as a homogenous nation, and during PCA analysis, the majority of the Eigenvalues tend to concentrate on PC1 (see Appendix A). Due to this tendency, most GWASs utilizing KoGES data typically include only PC1 and PC2 as covariates. This highlights the importance of carefully accounting for population structure in GWASs involving Korean or other genetically homogenous populations, as such adjustments ensure robust and unbiased results. This study builds upon and expands the understanding of previously identified genetic loci in Asian populations, offering new perspectives on their role in hypertension. For example, previous studies have reported an association between hypertension and ATP2B1 [38,39] and ALDH2 [40,41] gene variants. In 2009, Hong et al. reported the most significant finding, namely, the rs17249754 variant of the ATP2B1 gene, which was associated with hypertension through a GWAS in 8842 Koreans [38]. Mei et al.’s 2020 meta-analysis on the ALDH2 rs671 polymorphism and its association with essential hypertension included 12 studies, all conducted in Asian populations. This is explicitly stated in the meta-analysis, which notes that all the included studies originated from Asia. Additionally, the authors highlight that the research focused on Asian cohorts, with no studies from European or other non-Asian populations included in the analysis. This geographical focus ensures the findings are specific to the genetic and environmental contexts of Asian populations [41]. In contrast, certain loci, such as rs7556898 (chr2q24.3) and rs1980854 (chr20p12.2), have not been previously reported in GWASs focused on hypertension, either in Asian or European cohorts. The identification of these loci in this study underscores the potential of metabolic risk stratification (e.g., METS-IR) in uncovering genetic variants that may otherwise remain undetected. Specifically, rs1980854, which has been associated with osteoporosis in prior studies, appears to influence cardiovascular risk in individuals with a high METS-IR, providing novel insights into its broader implications across different populations [42,43,44]. In this study, the rs1980854 (chr20p12.2) variant was shown to increase prevalence, even in the high-METS-IR population, suggesting a new interpretation as a notable genetic variant affecting the cardiovascular system. These findings suggest that while some loci show consistent associations across populations, others may exhibit population-specific or metabolic context-dependent effects. Further studies are warranted to validate these novel loci and explore their functional roles in diverse populations.

These results indicate the necessity for more rigorous management strategies for individuals with this genetic mutation who are at high prevalence for hypertension. Furthermore, these findings provide crucial evidence for the development of personalized treatment strategies. In particular, patients with high METS-IR levels and those with the rs1980854 (chr20p12.2) mutation may require more active hypertension prevention strategies. Such personalized approaches may lead to superior clinical outcomes in the management of hypertension.

This study has notable strengths in its analysis of the interactions among hypertension prevalence, METS-IR, and genetic variations. This study yielded reliable results through the analysis of large-scale GWAS data and the application of multivariate statistical analysis. Of particular significance is the evaluation of the role of genetic predispositions in East Asian populations, focusing on the Korean population. Nevertheless, the study is limited by the difficulty in establishing a causal relationship because of the observational nature of the study design. Additional studies that incorporate diverse racial groups and environmental factors are necessary to enhance the generalizability of these findings. Additionally, the precise effect of specific SNPs remains unclear, necessitating further functional studies to elucidate their mechanisms of action.

The results of this study indicate that a comprehensive approach that considers both genetic and metabolic factors is necessary for the effective management of hypertension. It will be crucial for future studies to assess whether these genetic variations are consistently expressed across different populations. Additional research is required to elucidate the mechanisms through which genetic variations contribute to the development of hypertension at the molecular level. This will ultimately facilitate the development of more effective strategies for the prevention and management of hypertension.

## 4. Materials and Methods

### 4.1. Study Population

Study participants were recruited from the Korean Genome and Epidemiology Study (KoGES) cohort, a genome-based epidemiological study funded by the Korean government, including the National Research Institute of Health, the Centers for Disease Control and Prevention, and the Ministry of Health and Welfare [45]. The primary objective of the KoGES was to investigate the genetic and environmental causes of common complex diseases in the Korean population [45]. The KoGES cohort comprises community residents and individuals recruited from the National Health Examinee Registry, including men and women aged 40 years or older at baseline [45]. The dataset contained comprehensive information on participants’ medical and pharmacological histories, anthropometric measurements, and blood biochemistry profiles [45]. Both prevalent and incident hypertension cases were included in this study. Prevalent cases were identified based on participants’ self-reported history of hypertension diagnosis or documented use of antihypertensive medications. A total of 72,299 participants with available genome-wide single nucleotide polymorphism (SNP) genotype data were included in the KoGES dataset. Informed consent was obtained from all participants before their inclusion in the study [45]. This study was performed in accordance with the Declaration of Helsinki and approved by the Institutional Review Board (IRB) of Dong-A University Hospital Institutional Review Board (DAUHIRB-EXP-24-138).

### 4.2. METS-IR Calculation and Tertile Grouping

The METS-IR was calculated using the following formula: ln [(2 × FBG + TG) × BMI]/ ln [HDL-C]. This score was categorized into tertiles for the analysis. The division into tertiles was chosen to balance granularity and statistical power, ensuring sufficient differentiation among low-, moderate-, and high-metabolic-risk groups while maintaining robust sample sizes. A binary division would oversimplify metabolic risk, and dividing into more than three groups could reduce subgroup sizes and statistical significance. This approach effectively captures group-specific genetic effects and provides clearer insights into the interaction between metabolic risk and hypertension. For our analysis, we limited the sample to participants without a history of cancer or thyroid disease to minimize the potential confounding effects related to hypertension. Furthermore, we excluded 1951 individuals with missing data on the METS-IR index and other covariates. The remaining participants were categorized into three groups based on their METS-IR tertiles. For this study, we selected participants from the first tertile (T1), defined as METS-IR < 34.22 in males and <30.94 in females, and the third tertile (T3), defined as METS-IR ≥ 39.25 in males and ≥35.93 in females. A total of 40,148 participants were included in the final analysis. This cohort comprised 3517 participants with hypertension, 16,240 without hypertension at T1, 8653 with hypertension, and 11,711 without hypertension at T3. A flowchart of the study population is shown in Figure 1.

### 4.3. Hypertension Definition

Hypertension was defined according to the following criteria: SBP ≥ 140 mm Hg, DBP ≥ 90 mm Hg, current use of antihypertensive medication, or a previous diagnosis of hypertension. Blood pressure measurements were obtained using a validated automatic sphygmomanometer after a 5 min rest period.

### 4.4. Genetic Data Collection

#### Genotyping

Fasting blood samples were collected in a serum separator tube and two ethylenediaminetetraacetic acid (EDTA) tubes. DNA was extracted from blood samples and sent to the National Biobank of Korea for analysis. SNP genotyping was performed using the Korea Biobank Array or KoreanChip (Axiom^TM^ KORV1.1, catalog ID: 550796, ThermoFisher, Singapore, Singapore), a tool specifically designed for GWASs of blood biochemical traits in the Korean population. KoreanChip includes over 833,000 markers, incorporating more than 247,000 rare or functional variants identified from the sequencing of over 2500 Koreans [46]. Detailed information on the Korean chip can be found in a previous study [46]. To ensure high-quality genotyping results, strict criteria were applied, including a call rate exceeding 97%, a missing genotype rate below 0.01, a minor allele frequency greater than 0.01, and a Hardy–Weinberg equilibrium *p*-value exceeding 0.000001.

### 4.5. Statistical Analysis

Logistic regression analyses were performed separately for the T1 and T3 groups to assess genetic associations with the prevalence of hypertension, adjusting for age, sex, smoking, alcohol consumption, physical activity, and the first two principal components (PC1 and PC2). High-throughput genome-wide analyses were conducted using PLINK (version 1.9.0), with the genome-wide significance set at *p* < 5 × 10^−8^. A heterogeneity test was performed to compare the effect size between the high- and low-METS-IR groups using PLINK (version 1.9.0). Heterogeneity was assessed using I^2^ and categorized into four groups: 0 to 40%, indicating weak heterogeneity; 30 to 60%, suggesting moderate heterogeneity; 50 to 90%, reflecting substantial heterogeneity; and 75 to 100%, representing considerable heterogeneity [47]. A Manhattan plot was generated using R (version 4.1.2; https://cran.r-project.org/bin/windows/base/, accessed on 11 June 2024) to visualize the GWAS results, and a Miami plot was used to compare outcomes between the two genome-wide association analyses. To further investigate the loci, a regional association plot was generated using LocusZoom (version 0.4.8.2) to confirm the surrounding SNPs and nearby genes in linkage disequilibrium with the lead SNPs. After identifying the lead SNPs, we selected unique SNPs that were significant only in T1 or T3, as well as those that were significant in both groups.

## 5. Conclusions

This study offers a comprehensive examination of the genetic interactions between hypertension and metabolic syndrome, indicating the potential for a novel personalized approach to the prevention and management of hypertension. Further investigations may substantiate these findings in diverse populations and elucidate the underlying functional mechanisms that could ultimately lead to more effective hypertension prevention and treatment strategies.

## Figures and Tables

**Figure 1 ijms-25-12690-f001:**
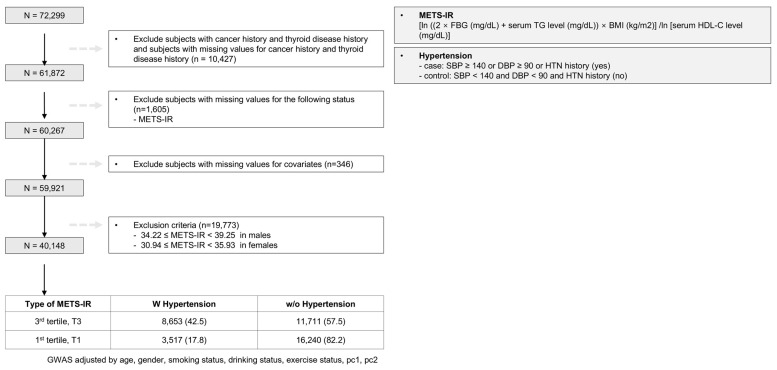
Study design overview.

**Figure 2 ijms-25-12690-f002:**
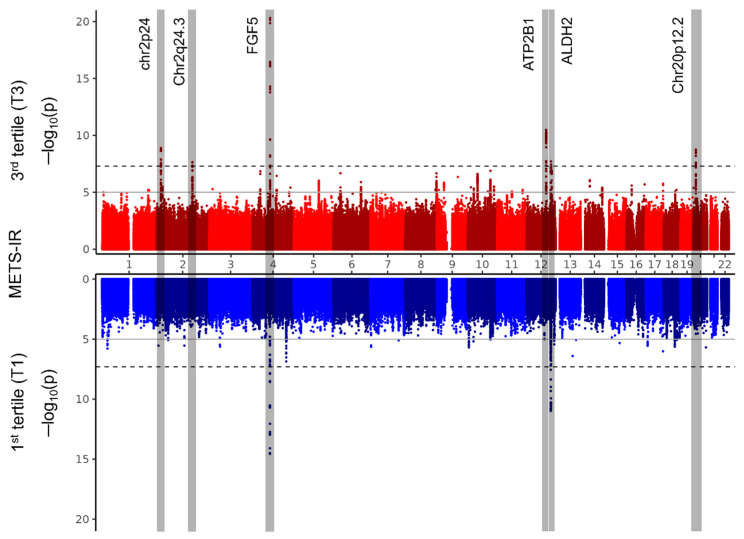
Miami plots. The results of the genome-wide association study (GWAS) analyzing the risk of hypertension prevalence in the first tertile (T1, blue graph) and third tertile (T3, red graph) groups based on METS-IR are presented in the Miami plots. The horizontal axis represents the position of SNPs across chromosomes 1 to 22, and the vertical axis shows the −log10 transformation of the *p*-values for hypertension associations. The solid line marks the genome-wide suggestive significance threshold (*p*-value < 1 × 10^−5^), and the dotted line indicates the genome-wide significance threshold (*p*-value < 5 × 10^−8^). Four regions (chr2p24, chr2q24.3, ATP2B1, and chr20p12.2) highlighted in gray show significant associations specific to the METS-IR T3 group, whereas the FGF5 and ALDH2 regions exhibit significant associations in both the T1 and T3 groups.

**Table 1 ijms-25-12690-t001:** Population characteristics.

	Total	METS-IR		
		1st Tertile, T1	2nd Tertile, T2	3rd Tertile, T3
N	59,921	19,774	19,773	20,374
Female (n, %)	37,455 (62.51)	12,360 (62.51)	12,360 (62.51)	12,735 (62.51)
AGE (mean ± sd)	53.57 ± 8.1	52.24 ± 8.23	53.84 ± 7.91	54.61 ± 7.96
Alcohol intake (g/week)	124.71 ± 235.83	116.03 ± 185.08	121.03 ± 283.76	138.33 ± 230.21
Lifestyle				
Smoking status (n, %): Never/ Quit/	42,534 (70.98)/ 9715 (16.21)/ 7672 (12.8)	14,193 (71.78)/ 3061 (15.48)/ 2520 (12.74)	14,176 (71.69)/ 3234 (16.36)/ 2363 (11.95)	14,165 (69.52)/ 3420 (16.79)/ 2789 (13.69)
Age of smoking onset (mean ± sd)	22.33 ± 5.83	22.32 ± 5.36	22.24 ± 5.74	22.43 ± 6.31
Duration of smoking (years, mean ± sd)	24.37 ± 10.83	24.05 ± 11.20	24.33 ± 10.79	24.68 ± 10.50
Drinking status (n, %): Never/ Quit/ Current	30,087 (50.21)/ 2252 (3.76)/ 27,582 (46.03)	9385 (47.46)/ 636 (3.22)/ 9753 (49.32)	9885 (49.99)/ 722 (3.65)/ 9166 (46.36)	10,817 (53.09)/ 894 (4.39)/ 8663 (42.52)
Exercise status (n, %): No/ Yes	29,128 (48.61)/ 30,793 (51.39)	9365 (47.36)/ 10,409 (52.64)	9228 (46.67)/ 10,545 (53.33)	10,535 (51.71)/ 9839 (48.29)
Disease				
Hypertension (n, %)	17,794 (29.72)	3517 (17.8)	5624 (28.46)	8653 (42.49)
Anthropometric traits				
Body mass index (kg/m^2^)	24.02 ± 2.91	21.37 ± 1.68	23.88 ± 1.52	26.72 ± 2.4
Systolic blood pressure (mmHg)	122.25 ± 15.31	118.04 ± 14.51	122.32 ± 14.86	126.25 ± 15.42
Diastolic blood pressure (mmHg)	75.93 ± 10.05	73.47 ± 9.66	75.88 ± 9.78	78.36 ± 10.1
Biochemical traits				
Fasting plasma glucose (mg/dL)	95.02 ± 19.95	89.84 ± 12.95	93.95 ± 16.91	101.08 ± 25.81
Total twoholesterol (mg/dL)	197.75 ± 35.66	194.8 ± 33.73	198.45 ± 35.56	199.93 ± 37.36
HDL cholesterol (mg/dL)	52.89 ± 13.1	62.22 ± 12.9	52.18 ± 10.27	44.53 ± 9.28
Triglyceride (mg/dL)	128.95 ± 89.6	84.36 ± 39.27	118.09 ± 58.09	182.76 ± 117.89

Lifestyle factors, disease prevalence, anthropometric measurements, and biochemical traits are shown across tertile groups based on the METS-IR. The data are presented as means ± standard deviations (SDs) or numbers (percentages).

**Table 2 ijms-25-12690-t002:** Lead SNPs of the genome-wide significant associations with hypertension prevalence for each METS-IR group.

SNP	CHR	BP	Locus	A1	Minor Allele Frequency				Association Result to Hypertension by METS-IR Type			** Heterogeneity	
					This Study	EAS	EUR	AMR	METS-IR	OR (95% CI)	* *p*	I^2^	Type
									Type				
rs11899121	2	20367973	chr2p24	C	0.36	0.41	0.46	0.5	** High **	** 0.87 (0.83–0.91) **	** 1.85 × 10^−9^ **	81.7	considerable
									Low	0.95 (0.90–1.01)	7.57 × 10^−2^		
rs7556898	2	165008513	chr2q24.3	T	0.43	0.4	0.33	0.57	** High **	** 0.89 (0.85–0.93) **	** 4.41 × 10^−8^ **	57.68	substantial
									Low	0.94 (0.89–0.99)	2.54 × 10^−2^		
rs10857147	4	81181072	FGF5	T	0.34	0.35	0.26	0.27	** High **	** 1.24 (1.19–1.30) **	** 4.84 × 10^−21^ **	0	weak
									** Low **	** 1.26 (1.19–1.33) **	** 8.02 × 10^−15^ **		
rs17249754	12	90060586	ATP2B1	A	0.38	0.31	0.14	0.11	** High **	** 0.86 (0.83–0.90) **	** 6.20 × 10^−11^ **	32.73	moderate
									Low	0.9 (0.85–0.96)	5.37 × 10^−4^		
rs671	12	112241766	ALDH2	A	0.16	0.17	0	0	High	0.85 (0.76–0.90)	1.91 × 10^−7^	80.56	considerable
									** Low **	** 0.75 (0.69–0.82) **	** 5.23 × 10^−11^ **		
rs1980854	20	10985803	chr20p12.2	A	0.27	0.32	0.28	0.13	** High **	** 1.15 (1.09–1.20) **	** 3.19 × 10^−8^ **	61	substantial
									Low	1.08 (1.02–1.15)	1.25 × 10^−2^		

SNP, single-nucleotide polymorphism; Chr, chromosome; BP, base pair; EAS, East Asian; EUR, European; AMR, American; OR, odds ratio; CI, confidence interval. * *p*-value was calculated using a logistic regression analysis adjusting for age, gender, smoking, alcohol consumption, physical activity, and the first two principal components (PC1 and PC2). ** Heterogeneity was assessed using I^2^ and categorized into four groups: 0 to 40%, indicating weak heterogeneity; 30 to 60%, suggesting moderate heterogeneity; 50 to 90%, reflecting substantial heterogeneity; and 75 to 100%, representing considerable heterogeneity.

## Data Availability

The datasets generated and/or analyzed in the current study are available from the corresponding author upon reasonable request. Requests to access these datasets should be directed to B.-K.S.

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
