# Peer review of "The Genetic Variants Influencing Hypertension Prevalence Based on the Risk of Insulin Resistance as Assessed Using the Metabolic Score for Insulin Resistance (METS-IR)"

_ijms, 2024, doi:10.3390/ijms252312690_

Round 1
Reviewer 1 Report
Comments and Suggestions for Authors
This study performed a GWAS for hypertension stratified by the Metabolic Score for Insulin Resistance, among Korean populations. The methods were not clearly described in the manuscript. It did not clearly describe whether prevalent cases were included in the analyses. If a large proportion of cases were prevalent cases, it should not use “risk of hypertension” across the manuscript. There are more major concerns as listed below.
Abstract, line 15: Change “genetic polymorphisms” to “genetic variants”. Same at line 23.
Abstract, line 16: Revise as “stratified by METS-IR score levels”.
Abstract, line 18: All abbreviations need definitions first, same for line 210 in the methods.
Abstract does not have discussion for the findings of this study. Lines 28-31 are too general conclusions.
Results, line 76: If the methods section is arranged at the end of the manuscript, a brief description of the study population is needed at the beginning of the results section, including the cohort’s name, race, and sample size.
Methods, line 195. It described the KoGES as a cohort study. Were prevalent hypertension cases at the baseline excluded? Did the analyses include incident cases only? Was it a nested case-control study in this manuscript? If so, which age for controls was used as a covariate? Age at baseline or last follow-up?
Methods, line 245: Why were only first two PCs included in the analyses? A clarification is needed.
Results, Table 2: A heterogeneity test should be performed to compare the effect size between METS-IR high and low groups. Then more discussions should be made focusing on those with significant heterogeneity.
Results, line 125: I do not agree with the descriptions of the results. GWAS identifies loci associated with risk of hypertension (BTW, if the analyses included lots of prevalent cases, it even cannot be described as “risk”). The minor allele of the leading SNP showed increased or decreased risk. We should not describe a locus that increases or decreases the risk. The results should be (1) description of loci found in METS-IR high group; (2) loci found in low group; (3) focus on loci showing association in one but not in the other group; (4) specify whether lead variants were located at coding region.
Discussion, line 152: As the second paragraph in the discussion, a clear description is needed to show how many/which loci identified in this study have not been reported by previous GWAS studies.
Discussion: Previous GWAS studies on the identified loci among Asian and European populations should be discussed.
Author Response
Response to Reviewer Comments
Manuscript ID: ijms-3297672
Manuscript title: Genetic Variants Influencing Hypertension Prevalence Based on the Risk of Insulin Resistance as Assessed by the Metabolic Score for Insulin Resistance (METS-IR)
Dear Editor,
Thank you for accepting and reviewing our manuscript for publication in the International Journal of Molecular Sciences (IJMS).
We sincerely appreciate the valuable comments and suggestions from the reviewers, which have helped us improve the quality of our work. Based on your feedback, we have thoroughly revised the manuscript and addressed each point carefully. The revised content is highlighted in blue in the updated manuscript for your convenience.
All authors have reviewed and approved the revised version of the manuscript, and we hope that it now meets the high standards of your esteemed journal. We appreciate the opportunity to submit our work to IJMS and appreciate your consideration for publication.
Reviewer 1 Comments
This study performed a GWAS for hypertension stratified by the Metabolic Score for Insulin Resistance, among Korean populations. The methods were not clearly described in the manuscript. It did not clearly describe whether prevalent cases were included in the analyses. If a large proportion of cases were prevalent cases, it should not use “risk of hypertension” across the manuscript. There are more major concerns as listed below.
Response: We sincerely appreciate the reviewer’s valuable comments. We also agree that the expression "risk of hypertension" is not appropriate for this study. Accordingly, we have revised all instances to "prevalence of hypertension."
Abstract, line 15: Change “genetic polymorphisms” to “genetic variants”. Same at line 23.
Response: We have revised "genetic polymorphisms" in Abstract, Line 15 to "genetic variants." However, in Line 23, the term "polymorphisms" was not changed to "variants" because it is part of the full description of SNPs (Single Nucleotide Polymorphisms). Instead, we added the abbreviation "SNPs."
Abstract, line 16: Revise as “stratified by METS-IR score levels”.
Response: The corresponding text has been revised.
Abstract, line 18: All abbreviations need definitions first, same for line 210 in the methods.
Response: we described the all abbreviations with the definition at the first use.
Abstract does not have discussion for the findings of this study. Lines 28-31 are too general conclusions.
Response: As per the reviewer’s suggestion, we have added further discussion of our results in the Abstract.
Results, line 76: If the methods section is arranged at the end of the manuscript, a brief description of the study population is needed at the beginning of the results section, including the cohort’s name, race, and sample size.
Response: As per the reviewer’s suggestion, we have described the cohort's name, race, and sample size at the beginning of the Results section as follows.
Methods, line 195. It described the KoGES as a cohort study. Were prevalent hypertension cases at the baseline excluded? Did the analyses include incident cases only? Was it a nested case-control study in this manuscript? If so, which age for controls was used as a covariate? Age at baseline or last follow-up?
Response: This study is not about incidence cases, and we agree with the reviewer’s comment that the term "Hypertension risk" in the context of METS-IR stratification could cause confusion. Therefore, we have revised the manuscript to replace all instances of "Hypertension risk" with "Prevalence of Hypertension."
Methods, line 245: Why were only first two PCs included in the analyses? A clarification is needed.
Response: All participants in this study were of Korean ethnicity. When analyzing the genome data of Koreans, the Eigenvalues for PC1 and PC2 reached 1, so no additional PCs were used as covariates. This approach to applying covariates is consistent with previously published studies based on the KoGES dataset.
Results, Table 2: A heterogeneity test should be performed to compare the effect size between METS-IR high and low groups. Then more discussions should be made focusing on those with significant heterogeneity.
Response: We conducted the heterogeneity test suggested by the reviewer and added the results to Table 2. The method used for the test is described on Page 8, Lines 285–290, and the related references have been provided on Page 10, Lines 430–431.
Results, line 125: I do not agree with the descriptions of the results. GWAS identifies loci associated with risk of hypertension (BTW, if the analyses included lots of prevalent cases, it even cannot be described as “risk”). The minor allele of the leading SNP showed increased or decreased risk. We should not describe a locus that increases or decreases the risk. The results should be (1) description of loci found in METS-IR high group; (2) loci found in low group; (3) focus on loci showing association in one but not in the other group; (4) specify whether lead variants were located at coding region.
Response: Following the reviewer’s suggestion regarding the presentation of results, we have revised the description on Page 5, Lines 135–143 as follows.
Discussion, line 152: As the second paragraph in the discussion, a clear description is needed to show how many/which loci identified in this study have not been reported by previous GWAS studies.
Response: As per the reviewer’s suggestion, we have included the description on Page 5, Lines 153–155.
Discussion: Previous GWAS studies on the identified loci among Asian and European populations should be discussed.
Response: We have described the results of previous GWAS studies between Page 6, Lines 171–196.

Reviewer 2 Report
Comments and Suggestions for Authors
Thanks for the opportunity to review this manuscript. Next, some considerations that I would like the authors to attend.
Minor comment: In the last paragraph of the introduction, remove "in conclusion" from the "objective" sentence.
In Table 1, please break down the information about the tobacco index, years of smoking, and age of onset of smoking. Consider that many previous research have described the participation of genes/SNPs in nicotine addiction.
Lines 121-123 read: "Although there were no functional genes within these regions, this suggests the presence of a preventive mechanism in our genome that can mitigate the risk of hypertension in individuals with a high risk of metabolic syndrome." These sentences are so speculative that they should be reformulated since no references support this subjective interpretation.
In general, the discussion section is scanty and vague. Most of the information that could be discussed is previously exposed in the results section. I recommend moving most of this information to the discussion and deep into the description of the results. Also, consider explaining the ancestral contribution using a PCA.
Please briefly describe (section 4.2) why the results were divided into tertiles for the analysis. What is the advantage of classifying like this?
In "Figure 2. Study Design Overview," remove "KoGES" from each cell; since all participants are from the same cohort, this is redundant.
In addition to reference 46, include the appropriate commercial information for KoreanChip, which includes a trademark, catalog and/or ID, source, city, and country.
Author Response
Response to Reviewer Comments
Manuscript ID: ijms-3297672
Manuscript title: Genetic Variants Influencing Hypertension Prevalence Based on the Risk of Insulin Resistance as Assessed by the Metabolic Score for Insulin Resistance (METS-IR)
Dear Editor,
Thank you for accepting and reviewing our manuscript for publication in the International Journal of Molecular Sciences (IJMS).
We sincerely appreciate the valuable comments and suggestions from the reviewers, which have helped us improve the quality of our work. Based on your feedback, we have thoroughly revised the manuscript and addressed each point carefully. The revised content is highlighted in blue in the updated manuscript for your convenience.
All authors have reviewed and approved the revised version of the manuscript, and we hope that it now meets the high standards of your esteemed journal. We appreciate the opportunity to submit our work to IJMS and appreciate your consideration for publication.
Reviewer 2 Comments
Thanks for the opportunity to review this manuscript. Next, some considerations that I would like the authors to attend.
Minor comment: In the last paragraph of the introduction, remove "in conclusion" from the "objective" sentence.
In Table 1, please break down the information about the tobacco index, years of smoking, and age of onset of smoking. Consider that many previous research have described the participation of genes/SNPs in nicotine addiction.
Response: As per the reviewer’s comments, we have added information on smoking years and the age of onset of smoking to Table 1.
Lines 121-123 read: "Although there were no functional genes within these regions, this suggests the presence of a preventive mechanism in our genome that can mitigate the risk of hypertension in individuals with a high risk of metabolic syndrome." These sentences are so speculative that they should be reformulated since no references support this subjective interpretation.
Response: Based on the reviewer’s feedback, we revised the Results section to include only objective findings. The speculative interpretation was removed from the Results and is now discussed appropriately in the Discussion section. In the Discussion, we noted that certain loci, such as rs7556898 (chr2q24.3) and rs1980854 (chr20p12.2), have not been previously reported in GWAS studies on hypertension in either Asian or European populations. This study highlights the potential of metabolic risk stratification (e.g., METS-IR) in identifying novel genetic variants that might otherwise go undetected. Revisions have been made to Lines 183–188 on Page 6.
In general, the discussion section is scanty and vague. Most of the information that could be discussed is previously exposed in the results section. I recommend moving most of this information to the discussion and deep into the description of the results. Also, consider explaining the ancestral contribution using a PCA.
Response: The participants in this study were all of Korean ethnicity. Korea is well-known as a homogenous nation, and during PCA analysis, the majority of the Eigenvalue tends to concentrate on PC1 (see Supplementary Figure 1). Due to this tendency, most GWAS studies utilizing KoGES data typically include only PC1 and PC2 as covariates.
Additionally, we have incorporated modifications to lines 165 through 171 of the discussion section to further elaborate on the novel findings and their implications.
Please briefly describe (section 4.2) why the results were divided into tertiles for the analysis. What is the advantage of classifying like this?
Response: Thank you for the important comments. We have added an explanation for the rationale behind analyzing by tertiles on Page 7, Lines 245–251, as follows.
In "Figure 2. Study Design Overview," remove "KoGES" from each cell; since all participants are from the same cohort, this is redundant.
Response: As per the reviewer’s comments, we revised the figure to remove "KoGES." Additionally, during the revision process, the figure number was changed to Figure 1.
In addition to reference 46, include the appropriate commercial information for KoreanChip, which includes a trademark, catalog and/or ID, source, city, and country.
Response: Thank you for your thorough review. The points you mentioned have been addressed and described as follows on Page 8, Line 272 –273.

Round 2
Reviewer 1 Report
Comments and Suggestions for Authors
All my concerns have been addressed.